# The Dual Role of Innate Immune Response in Acetaminophen-Induced Liver Injury

**DOI:** 10.3390/biology11071057

**Published:** 2022-07-14

**Authors:** Tao Yang, Han Wang, Xiao Wang, Jun Li, Longfeng Jiang

**Affiliations:** 1Department of Infectious Diseases, The First Affiliated Hospital with Nanjing Medical University, Nanjing 210029, China; yangtao@stu.njmu.edu.cn (T.Y.); wh19825802773@163.com (H.W.); wangxiao108@stu.njmu.edu.cn (X.W.); 2Department of Respiratory and Critical Care Medicine, The Affiliated People’s Hospital of Jiangsu University, The Zhenjiang Clinical Medical College of Nanjing Medical University, Zhenjiang 212001, China

**Keywords:** innate immune response, acetyl-para-aminophenol-induced liver injury, neutrophils, macrophages, cytokine

## Abstract

**Simple Summary:**

An injury to the liver caused by a drug or its metabolites is referred to as drug-induced liver injury (DILI). The clinical manifestations of DILI are varied, and it may even result in acute liver failure under certain circumstances. Based on findings in the United States and China, Acetyl-para-aminophenol (APAP) may not be the most prevalent cause of DILI, but it is the leading cause of acute liver failure. The study of immune responses in APAP-induced liver injury (AILI) has been making significant progress in recent years. It should be noted, however, that different studies have reported differing results regarding the role played by immune cells in AILI, with some studies indicating they are proinflammatory, while others have showed no effect, or even pro-repair effects. Therefore, we here discuss the mechanisms of the dual role of immune cells in AILI.

**Abstract:**

Acetyl-para-aminophenol (APAP), a commonly used antipyretic analgesic, is becoming increasingly toxic to the liver, resulting in a high rate of acute hepatic failure in Europe and the United States. Excessive APAP metabolism in the liver develops an APAP–protein adduct, which causes oxidative stress, MPTP opening, and hepatic necrosis. HMGB-1, HSP, nDNA, mtDNA, uric acid, and ATP are DMAPs released during hepatic necrosis. DMAPs attach to TLR4-expressing immune cells such KCs, macrophages, and NK cells, activating them and causing them to secrete cytokines. Immune cells and their secreted cytokines have been demonstrated to have a dual function in acetaminophen-induced liver injury (AILI), with a role in either proinflammation or pro-regeneration, resulting in contradicting findings and some research confusion. Neutrophils, KCs, MoMFs, NK/NKT cells, γδT cells, DCs, and inflammasomes have pivotal roles in AILI. In this review, we summarize the dual role of innate immune cells involved in AILI and illustrate how these cells initiate innate immune responses that lead to persistent inflammation and liver damage. We also discuss the contradictory findings in the literature and possible protocols for better understanding the molecular regulatory mechanisms of AILI.

## 1. Introduction

Drug-induced liver injury (DILI) is a common hepatology disease caused by the direct or indirect action of the drugs or their metabolites on the liver after exposure to a specific drug in clinical practice. It can present in various presentations, ranging from asymptomatic to hepatocellular or cholestatic jaundice, liver failure, or chronic hepatitis [1]. The most prevalent medicines that induce DILI in Western countries include antimicrobials, nonsteroidal anti-inflammatory drugs (NSAIDs), herbal and dietary supplements (HDS), cardiovascular drugs, central nervous system (CNS) agents, and anticancer drugs. A prospective study from the United States showed that the annual incidence was 2.7 cases of DILI per 100,000 adults in 2014, with antibiotics (45.4%), HDS (16.1%), and cardiovascular drugs (10%) [2]. A multicenter, retrospective study from China included 25,927 patients with DILI in 308 medical centers. It showed that the annual incidence of DILI in the general population in mainland China was 23.8 per 100,000 persons, which is higher than the data reported in Western countries [3]. Traditional Chinese medicines or HDS (26.81%), antituberculosis medications (21.99%), and antineoplastic drugs or immunomodulators (8.34%) were the leading causes of DILI in China [3].

DILI is classified into two categories: intrinsic (direct) DILI is dose-and time-dependent and happens in a substantial proportion of people exposed to the medicine (predictable), occurs quickly (hours to days) after therapy, and can be replicated in animal models. Idiosyncratic DILI is frequently dose-independent, affects only a tiny fraction of people exposed (unpredictable), and manifests after a day-to-week delay; it is often unreproducible in animal models [4]. In addition, indirect DILI is a new and not yet entirely accepted category, defined as hepatotoxicity secondary to the biological effects of the drug [5], which is partially dose- and time-dependent, somewhat predictable, occurs over several months, and is partially reproducible in animal models. The represented drugs include immune checkpoints inhibitors, monoclonal antibodies (anti-CD20 rituximab), protein kinase inhibitors, anti-PSCK9 (hypercholesterolemia), daclixumab, and corticosteroids [5].

Acetyl-para-aminophenol (APAP), also known as paracetamol or N-acetyl-p-aminophenol, is likely the most well-known and extensively used drug to cause intrinsic DILI [6]. APAP is also not dose-dependent in some idiosyncratic individuals, suggesting that host susceptibility factors are involved in the pathogenesis of APAP-induced liver injury (AILI). According to studies in the United States and China, APAP may not be the most prevalent cause of DILI, but it is the leading cause of acute liver failure (ALF). APAP overdose was the leading cause of ALF and responsible for 46% of all ALF cases in the USA [7], ranking as the first etiology in California [8] and 42% of patients in Sweden [9]. AILI causes a fast-progressing condition that includes liver and kidney failure, hepatic encephalopathy, acidosis, and multi-organ failure (MOF) [10]. AILI has few therapeutic approaches, necessitating liver transplantation in severe cases, but few patients are eligible for a donor’s liver. As a result, AILI impairment has become a public health concern. Considering the complex mechanisms of immune responses in APAP hepatotoxicity, this study systematically reviews the role of innate immune cells, secreted cytokines, and secreted chemokines against the APAP challenge. It provides a fundamental basis for future research.

## 2. Metabolism of APAP

APAP is a non-steroidal antipyretic and analgesic medication, and the clinically safe dosage varies from 1 to 4 g per day. Following oral treatment, APAP is absorbed from the gut and transported to the liver for metabolism, which requires a variety of enzymatic reactions [11,12]. The bulk (80–90%) of APAP is metabolized by UDP glucuronosyltransferase (UGTs) and sulfate transferase (SULTs) to inactive glucuronide (APAP-gluc) and sulfate (APAP sulfate), which is then eliminated in the urine, blood and bile, commonly known as the phase II metabolism reaction [13]. The phase I enzymatic reaction occurs when a minor (5–10%) amount of APAP is metabolized in hepatocytes by cytochrome P450 (CYP450) enzymes, such as CYP2E1 and CYE1A2, to the reactive metabolite N-acetyl-p-benzoquinone imine (NAPQI). The hepatic antioxidant glutathione (GSH) rapidly converts NAPQI to a harmless form via glutathione-S-transferase in an enzymatic process, forming the APAP-GSH complex (APAP-GSH), which is further metabolized to N-acetyl-L-cysteine adducts, cysteamide adducts, and cysteamide/glycine adducts, excreted through the urine or bile [14].

When the phase II reaction is saturated, excessive APAP is metabolized by the phase I reaction, NAPQI accumulation (Figure 1). When GSH is depleted, the growing concentration of NAPQI forms harmful APAP protein adducts, also known as NAPQI–protein adduct peptides, by covalently reacting with protein sulfur groups. In conclusion, both the generation of APAP protein adducts and the depletion of GSH result in increased protein adduct formation in hepatocytes, resulting in oxidative stress and increased reactive oxygen species (ROS) production. However, in the mitochondria of hepatocytes, those protein adducts appear to trigger limited oxidative stress, which is responsible for the induction of mitogen-activated protein kinase (MAPK); then, MAPK kinases (MKK)-4/7 activate c-Jun N-terminal kinase (JNK), leading to phosphorylation of JNK in the cytosol [15]. Then, p-JNK is translocated to mitochondria, where it binds the outer mitochondrial membrane anchor protein Sab (SH3-domain-binding protein that preferentially associates with Btk) or Sh3bp5 [16], generating dephosphorylation of intermembrane Src [17]. Thus, activated JNK and mitochondria p-JNK translocation, in turn, further enhances oxidative stress signal to induce the mitochondrial membrane permeability transition pore (MPTP) opening, causing mitochondrial membrane permeability and dysfunction [18], ultimately leading to hepatocyte necrosis and liver failure. This process needs various kinases involved, for example, mitochondria activating MAP3 kinases (MAP3K) such as maxed-lineage kinase-3 (MLK-3), apoptosis signal-regulating kinase-1 (ASK-1), and MAP2K [19], receptor-interacting protein 1(RIP1), and RIP3 [20]. However, more research is needed to magnify our understanding of the comprehensive relationship between kinases and JNK signals.

Then, the necrotic hepatocyte subsequently releases various endogenous damage-associated molecular patterns (DAMPs) (Figure 1), including high mobility group box protein 1 (HMGB-1) [21], heat shock proteins (HSPs), nuclear DNA fragments (nDNA fragments) [22] and mitochondrial DNA (mtDNA), uric acid [23], adenosine triphosphate (ATP) [24], and so on, thereby activating the innate immune response. Furthermore, necrotic hepatocytes also upregulated the infiltration of leucocytes [25], monocytes, activated Kupffer cells (KCs), and hepatic stellate cells (HSCs) [26]. KCs and bone marrow-derived monocytes/macrophages recognize DMAPs via Toll-like receptor (TLR) 4 and are activated, releasing proinflammatory cytokines and chemokines, resulting in a “cytokines storm”. In particular, the HMGB-1 binds to and activates KCs via TLR4 [27]; activated KCs release mediators that directly trigger cell death, including tumor necrosis factor (TNF)-α, Fas ligand (FasL, CD95L), and ROS, or indirectly cause cell death through the recruitment of neutrophils, including interleukin (IL)-1β and CXCL2 [28]. Another critical step in generating an inflammatory response is the formation of inflammasomes, a multiprotein complex, in immune cells [29]. Despite these consequences, the pathophysiological relevance of this sterile inflammatory response following APAP is still a hot issue of debate.

## 3. The Dual Role of Immune Cells and Cytokines

### 3.1. Neutrophils in AILI

Neutrophils are the first-line defense of the innate immune response, which we view as the mighty capacity for biosynthetic activity, including complement components, Fc receptors, cytokines, and chemokines [30]. Neutrophils usually are inactivated and travel slowly and aimlessly through the peripheral blood circulation. When the pathogen invades or endogenous stimulants are released, pattern recognition receptors can identify pathogen-associated molecular patterns (PAMPs) and DAMPs. The neutrophils in circulation are activated, which migrate to the injury site [31]. The study revealed that the recruited neutrophils (Mac-1^+^ Gr-1^+^) significantly increased in the liver at 6 and 24 h after APAP treatment, and during the recovery period, neutrophils subsequently declined [32].

How do neutrophils recruit to the area of hepatocyte necrosis? First is the release of various endogenous DAMPs by necrotic hepatocytes, as demonstrated in liver biopsies [33]. The HMGB1-TLR4-IL-23-IL-17A axis can facilitate neutrophil penetration after APAP treatment, whereas the HMGB1 inhibitor glycyrrhizin drastically inhibits IL-23 and IL-17A production as well as hepatic neutrophil accumulation [34]. Further study by L et al. found that APAP increases HMGB1 expression via activating Caspase-1 in hepatocytes, while neutrophil depletion or abolishing neutrophil extracellular traps (NETs) formation reduces HMGB1 levels and prevents hepatocyte necrosis [35]. Therefore, those studies confirmed that HMGB1 is an indeed targeted DAMP. The partly humanized anti-HMGB1 monoclonal antibody (mAb; h2G7), as compared with NAC, has high therapeutic efficacy and an extended therapeutic window in APAP-ALI [36]. Other DAMPs, such as the release of endogenous ATP, activate the purinergic P2 receptor (P2R) [37] and uric acid [23], promoting neutrophil infiltration and subsequent mouse hepatocyte death; mtDNA activates neutrophils by binding to TLR-9 [38]. In addition to DAMPs, other molecules, such as CXC chemokine receptor 2 (CXCR2), formylated peptide [39], and intercellular adhesion molecule-1 (ICAM-1) [40] could rapidly attract neutrophils to sites of hepatocytes necrosis. The third is other immune cells, such as NK cells and NKT cells, regulating neutrophil accumulation by secreting INF-γ [41]. However, whether the recruited neutrophils have an injurious or protective role in AILI is controversial. 

It has been hypothesized that neutrophil infiltration protects against hepatic necrosis and promotes liver healing. Hepatic damage has been associated with neutrophil recruitment in studies. There was no increase in Mac-1 (CD11b/CD18) expression or L-selectin shedding on circulating neutrophils, and anti-CD18 antibodies were not protective against AILI in the first 24 h [42]. This neutrophil infiltration was dominated by removing necrotic cell debris but was not robust enough to cause further damage. Further studies found that CD18-deficient mice did not differ from wild-type mice regarding an inflammatory response or liver injury in AILI [43]. Neutrophil depletion by anti-neutrophil antibody Gr-1, gp91phox^−/−^ (an essential subunit of NADPH oxidase, a significant source of phagocytic superoxide) [44], anti-Ly6G antibody, genetic knockout in granulocyte colony-stimulating factor, or genetic deletion in NADPH oxidase 2 (Nox2) [45] did not protect against APAP hepatotoxicity. Those studies suggested that AILI is dominated by intracellular cell death mechanisms in mice rather than being neutrophil-mediated [43]. The possible reason is that neutrophils are not activated in the early stages of AILI and fail to induce injury. Instead, the activation status (CD11b expression and ROS priming) during and after the damage peak [44]. Then, the neutrophils promote the phenotypic conversion of proinflammatory Ly6C^hi^CX3CR1^lo^ monocytes/macrophages to pro-resolving Ly6C^lo^CX_3_CR1^hi^ macrophages by expressing reactive oxygen species (ROS), promoting liver repair after APAP injury [45] (Table 1).

In APAP overdose, it has also been suggested that neutrophils cause liver impairment. Liu et al. used an anti-Gr-1 antibody (RB6-8C5) to deplete neutrophils in vivo, which effectively protected against AILI in mice model through reduced FasL expression, direct hepatocytotoxicity, and mitochondrial respiratory chain burst (NO, iNOS) in hepatic leukocytes [32], moreover, ICAM-1 was also involved in the injury process, and AILI was significantly reduced in ICAM-1 deficient mouse. Another study blockade of neutrophil infiltration by anti-granulocyte receptor 1 depletion or combined CXCR2-FPR1 antagonism significantly prevented liver injury after APAP overdose [46]. Both studies used anti-neutrophil antibodies, which selectively deplete circulating neutrophils, causing most neutrophils to wedge in the capillary bed, where they were recognized and phagocytosed by KCs in the liver activated KCs. Activated KCs amplify the inflammatory response and exacerbate the injury by releasing cytokines through phagocytosis.

The precise mechanisms of action of neutrophils in AILI are challenging since the exact process by which circulating neutrophils are entirely depleted and interact with other immune cells is unknown. During the injury and recovery stages of AILI, the infiltrating Ly6C^hi^ monocytes, their macrophage descendants, and neutrophils spatially and temporally overlap in the centrilobular necrotic areas [47]. A decrease in neutrophils that produce ROS has been observed following inducible ablation of circulating Ly6C^hi^ monocytes [47]. On the other hand, these controversial opinions may be associated with the experimental protocol of the researcher. Anti-CD18 antibodies, another example, were previously shown to reduce neutrophils by only 50% in an endotoxin shock model [48], which could explain why mice treated with anti-CD18 antibodies are not protected against AILI; similarly, CD18-deficient, neutrophil-specific antibody Ly6G or congenital neutropenia did not affect the AILI. Because partial or selective inhibition may lead to debatable findings, complete eradication of neutrophils and their function is required to account for the crucial role of neutrophils in AILI [49]. Lastly, the experimental observation time point may be relevant. AILI is a progressive process; for example, mechanisms of injury caused by different immune cells recruited at other time points may diverge, so investigators may need to study injury mechanisms at several different time points within 24 h and continually evaluate them.

**Table 1 biology-11-01057-t001:** The dual role of immune cells in AILI.

	Proinflammatory	No Effect or Pro-Regenerative
Neutrophils	Depleted neutrophils could protect against AILI via reduced FasL-expression, hepatocytotoxicity, and mitochondrial respiratory chain burst [32]. Blockade of neutrophil infiltration by anti-granulocyte receptor 1 depletion or combined CXCR2-FPR1 antagonism prevented liver injury [46]	No activation of circulating and liver neutrophils during AILI [42]. Neutrophil infiltration could be moving necrotic cell debris but not cause further damage, and CD-18-deficient mice were not protected [43]. gp91phox^−/−^ did not protect [44]. Anti-Ly6G, genetic knockout in granulocyte-colony-stimulating factor, or genetic deletion in Nox2 did not protect against APAP overdose, promoting the phenotypic conversion of proinflammatory macrophages to pro-resolving macrophages, and promoting liver repair [45]
KCs	Depletion of KCs can restrain APAP injury [50,51]. Mincle deletion (or KCs depletion) may reduce APAP hepatotoxicity [52].	EPO promotes the proliferation and function of KCs, ameliorating AILI [53]. Depletion of KCs can lead to liver injury aggravation [54,55,56]. KCs against AILI by secreting cytokines [57,58,59,60]
MoMFs	The activated MoMFs produce O2.-, NO., and peroxynitrite, promoting AILI progression [51], and upregulating proinflammatory gene expressions [61,62].	Upregulate endocytosis- and apoptotic-cell-clearance-related proteins which promote liver repair [63]. Promotes macrophage differentiation [64,65,66,67].
DCs		Prevent NKs cell activation and induce neutrophil apoptosis to reveal a protective role [68].
NK/NKT cells	Amplified the immune response, upregulated proinflammatory cytokine expressions, and increased neutrophil accumulation [41]. DMSO activated NK/NKT cells [69]. NKT-cell-deficient (Jα18^−/−^) mice could be resistant to AILI [70].	NKT cell-deficient mice (CD1d^−/−^ and Jα18^−/−^) were more vulnerable to AILI [71]. Reduce the release of inflammatory cytokines [72].
γδT cells	Depletion of γδT cells reduced IL-17A production and attenuated liver injury [34]. HIF attenuated abnormal γδT cell recruitment and alleviated AILI [73].	

HIF—hypoxia-inducible factor-1; Nox2—NADPH oxidase 2; APAP—acetyl-para-aminophenol; AILI—APAP-induced liver injury; EPO—erythropoietin; KCs—Kupffer cells; MoMFs—monocyte-derived macrophages; DCs—dendritic cells; NK/NKT cells—natural killer cells and NKT cells; DMSO—dimethyl sulfoxide; IL—interleukin.

### 3.2. Macrophages in AILI

Hepatic macrophages belong to the mononuclear macrophage system and respond to various liver injury signals. The macrophages observed in the currently damaged liver are heterogeneous and have two primary sources: liver-resident macrophages, named Kupffer cells (KCs), which are long-lived, self-renewing, and non-migratory macrophages [74]; the other source is blood/bone marrow monocyte-derived macrophages (MoMFs) and peritoneum macrophages, which identify danger signals, such as cytokines and chemokines, migrating to the liver. Liver macrophages that originate from various sources differ in activation and function, with a remarkably flexible and phenotypic alteration. Their functions are multidimensional and even opposing, directly influencing the outcome of the immune response [75].

#### 3.2.1. Kupffer Cells (KCs)

KCs are guardians within the liver sinusoid, constituting 90% of all tissue macrophages derived from yolk sac-derived progenitor cells [76]. KCs mainly express CD11b^+^ F4/80^++^CD68^+^CD11c^+/−^CLEC4F^+^TIM4^+^ and TLR4, TLR9, and CRIg, while not expressing chemokine receptor CX3CR1 in murine models of liver injury [77]. Human KCs express CD68^+^CD32^+^MARCO^+^TIMD4^+^ for identification [78]. KCs have a strong phagocytic ability to recognize and remove exogenous substances such as cell debris, pathogens, or apoptotic cells [49]. When pattern recognition receptors such as TLRs activate the KCs, pro- and anti-inflammatory cytokines, chemokines, and other molecules are released [79]. In APAP hepatotoxicity, the KCs have been protective and harmful in different studies [50,54].

In APAP-ALF, resident 51.5% of KCs were activated [80]. Therefore, KCs are the main subsets of phagocytosis in the liver at the onset of the injury. While recruited macrophages usually appear after 48 h, these resident KCs can initiate the specific response within this time window [78]. Previous studies reported that the number of KCs was reduced in the early stages of the AILI mouse model [28], and enhancing the activity of KCs could protect against liver injury. For example, erythropoietin (EPO) stimulated the proliferation of KCs and enhanced their phagocytosis, thereby ameliorating liver injury [53]. Moreover, the copper metabolism MURR1 domain (COMMD)10 appears indispensable for KCs survival [81]. On the other hand, depletion of KCs can lead to liver injury aggravation. Ju et al. used macrophage depleted agents (liposome/clodronate) to eliminate KCs, resulting in a significant decrease in the levels of hepatic mRNA expression of hepato-regulatory cytokines and mediators, such as IL-6, IL-10, IL-18 binding protein, and complement 1q, promoting exacerbation of AILI [55]. Similar results have been seen in other studies, and the mechanism may be related to the upregulation of multidrug resistance-associated protein 4 [54]. Other studies suggest a protective effect of KCs against AILI, which due to the production of cytokines [57], including the hypoxia-inducible factor (HIF)-2α, is reprogrammed in KCs to produce IL-6 [58]; blocking IL-33 increased liver injury by consuming KCs and boosting hepatic inflammatory factor releases, such as TNF-α, IL-6, and IL-1 [59]. There are also chemokines involved in KCs to promote the repair process in the liver, such as CCL2, CCL5, and CXCR2. CXCR2 was explicitly upregulated in hepatocytes around the necrotic region 24 h after APAP treatment. While through the secretion of IL-10, KCs can influence CXCR2 expression and pro-regenerative gene expression in surviving hepatocytes. Thus, KCs assist hepatocytes in shifting near the necrosis region to a proliferative state [60]. These studies suggest the resident KCs could prevent APAP impairment and promote liver restoration through secreted cytokines and chemokines.

However, many studies indicated that KCs promote APAP hepatotoxicity. One study concluded that KCs are a significant source of peroxynitrite. Blocking the formation of peroxynitrite using gadolinium chloride (GdCI3, KCs depleted agents) could restrain APAP injury; thus, the researchers concluded that KCs promote AILI through the release of peroxynitrite and oxidant stress [51]. Another study found that depletion of KCs using GdCI3 and clodronate in an AILI model resulted in downregulation of proinflammatory factor gene expression, improved liver injury, and significantly higher survival rates in mice [50]. Blocking TLR4 (using TLR4 antagonists and TLR4 mutants) can produce similar outcomes as depleting KCs [50]. However, other studies did not show a protective effect against AILI after using GdCI3 in KCs [55,56]. Macrophage-inducible C-type lectin (Mincle) produced from KCs was discovered in a recent study. The liver insult in Mincle KO mice was decreased after APAP treatment, as evidenced by reduced histological damage, cytokine downregulation, and neutrophil infiltration. Mincle’s harmful effects might be eliminated if KCs cells are removed. As a result, Mincle deletion (or KCs depletion) may reduce APAP’s hepatotoxicity [52].

Although it is known that KCs are decreased during the early stages of AILI, this reduction is unknown. First, determine if a direct mechanism of APAP toxicity causes KCs death or whether there is an antigenic switching on the surface of KCs (e.g., the disappearance of F4/80). Second, KCs function changes, such as researchers observed functional inactivation in KCs following GdCI3 treatment in animal models [82]. Therefore, KCs were decreased in quantity and functionally inactivated in AILI. 

Why do studies using the same KCs depleted agents (GdCI3 or liposome/clodronate) lead to different results in AILI? The proinflammation and pro-reparation KCs may be related to an investigator’s study protocol and the KCs phenotype. To begin with, KCs are heterogeneous in origin, phenotype, and function. For example, through single-cell sequencing, Blériot et al. found that classical KCs could be divided into KC1 and KC2, and *Esam* was the highest differentially expressed gene between the two groups. Functionally, KC2 was more involved in lipid metabolism [83]. Second, different studies may have different results in selecting KCs depleted agents, such as GdCI3, which has not been proved to have a protective effect in other trials, and the liposome/clodronate, which may cause amplified liver injury [55]. Moreover, in early injury, KCs appear to be decreased in quantity and become functionally inactive; therefore, utilizing medicines to deplete KCs is pointless. Third, the absence of NADPH oxidase activity is a significant source of superoxide in all phagocytes, choosing an appropriate inhibitor (NADPH oxidase deficit) to eliminate KCs as a source of oxidant stress may be more persuasive [44]. Finally, the relevant role of immune cells in the organism is complex, as using MCOX-E36 (a CCR2 inhibitor) can also reduce Ly6C^hi^ cells infiltration and mitigate AILI [28].

#### 3.2.2. Monocyte-Derived Macrophages (MoMFs)

Macrophages are plastic and diverse. In mice, bone marrow-derived macrophages highly express Ly6C and CCR2 (Ly6C^hi^CD11b^+^CCR2^++^CX3CR1^+^CD14^++^CD16^+/−^), whereas spleen-derived Ly6C^lo^ monocytes (Ly6C^lo^CD11b^+^CCR2^+^CX3CR1^++^CD14^−^CD16^+^) extensively express scavenger receptor with phagocytosis of apoptotic cell debris [77]. In humans, Ly6C expression is indistinguishable, and monocytes are classified according to CD14 and CD16 expression into classical (CD14^++^CD16^−^), intermediate (CD14^+^CD16^+^), and non-classical (CD14^−^CD16^+^) subpopulations [84]. Human CD14^++^CD16^−^ monocytes resemble mouse Ly6C^hi^ macrophages, whereas CD14^−^CD16^+^ monocytes are closer to Ly6C^lo^ macrophages [84]. During liver injury, circulating monocytes are recruited to the site of liver injury by adhesion molecules and chemokines are secreted from liver sinusoidal endothelial cells (LSEC), such as the CCL2/CCR2 axis, the CCR8/CCL1 axis, the CCL25/CCR9 axis, and the CXCR3/CXCL10 axis [85,86]. When CCR2^+^Ly6C^hi^ monocytes are recruited to the wound sites to promote organ injury, they subsequently undergo functional conversion and differentiate into mature Ly6C^lo^ monocytes to promote injury repair. Application of transcriptomic analysis of MoMFs revealed that Ly6C-classified macrophages behaved differently from the classical M1 (proinflammatory) and M2 (pro-restorative) [87]. Firstly, macrophages receive mixed signals in local remission that change dynamically in time and space [88]. Secondly, macrophages behave diversely and can reversibly switch with each other [89]. Therefore, macrophages destroy tissues by secreting proinflammatory cytokines and producing reactive oxygen species, but at the same time they have anti-inflammatory properties and promote tissue repair. This phenotypic and functional diversity leads to a complex role of macrophages in AILI, which exhibits multidimensional characteristics.

For the first time, Holt et al. used flow cytometry analysis and specific fluorescent-labeled antibodies to identify APAP-induced macrophages (CD11b^hi^F4/80^lo^CX3CL1^+^CCR2^+^) derived from circulating monocytes infiltrating the liver [90]. The recruited MoMFs are relayed on the CCR2 and M-CFS to mediate [61], occurring within 12–24 h of liver injury [28]. The infiltrated MoMFs are the main population of liver macrophages, and the freshly infiltrated MoMFs were highly expressed in Ly6C. Subsequently, the activated MoMFs produce superoxide (O2.-), nitric oxide (NO.), and peroxynitrite, promoting AILI progression [51]. Another mechanism of MoMFs enabled liver injury occurs through the upregulation of proinflammatory gene expressions, such as TNF-α, IL-1β, IL-6, and vascular endothelial growth factor (VEGF) [61,62]. Ly6C^hi^ MoMFs gradually decrease after 24 h of injury, afterward differentiates into pro-regenerative Ly6C^lo^ MoMFs within 48–72 h, and Ly6C^hi^ MoMFs almost disappear during the regeneration phase [61]. The neutrophils participate in the phenotypic conversion of Ly6C^hi^ MoMFs [45]. 

Ly6C^hi^ and Ly6C^lo^ MoMFs exist in different stages of AILI and have various tasks. The former regulates innate immune function and neutrophil survival, while the latter is responsible for neutrophil clearance [47]. Further proteomics analysis of Ly6C^lo^CX3CR1^hi^ MoMFs showed that upregulation of endocytosis- and apoptotic-cell-clearance-related proteins were involved in liver repair and regeneration [63]. The administration of macrophage-colony-stimulating factor (CSF-1) in mouse AILI can promote the differentiation of infiltrating macrophages, restore innate immunity in the liver, and accelerate recovery [64]. Mer tyrosine kinase (MerTK) is a phagocytic receptor that recognizes apoptotic cells and promotes the functional conversion of proinflammatory macrophages to pro-resolution macrophages [65]. The number of hepatic MerTK^+^MHCII^hi^ macrophages was significantly increased in experiments in the applied macrophage treatment AILI mouse model, while Mer^−/−^ mice exhibited persistent liver injury and inflammation. The mechanism may be that the secreted leukocyte protease inhibitor (SLPI) induces a MerTK^+^MHCII^hi^ phenotype that promotes apoptosis and clearance of neutrophils, reprogramming myeloid cells towards a resolution of the response and boosting liver injury reconstruction [66]. Downregulation of the proinflammatory sensor NLRP3 and overexpression of anti-inflammatory IL-10, which reinforces a pro-phagocytic phenotype via downstream STAT3 signaling and autocrine IL-6 stimulation, mediates a phenotypic shift from proinflammatory and pro-resolution macrophages via phagocytosis of debris by macrophages and interaction with neutrophil in AILI [67].

Currently, some cell-based therapies are also being actively explored in the experimental model of AILI. For instance, in an animal model of AILI, injection of activated syngeneic primary myeloid macrophages (AAMs) reduced hepatocyte necrosis, HMGB1 translocation, and neutrophil infiltration, and decreased levels of circulating proinflammatory factors, which stimulated the proliferation of hepatocytes and endothelial cells. The mechanism may be related to reducing host Ly6C^hi^ macrophages, demonstrating that AAMs are crosstalk with the host’s innate immune response [91].

These studies provide new perspectives on the proinflammatory and pro-regenerative macrophages in AILI. Whereas macrophages have different phenotypes at different stages and thus play different roles, the transition of macrophages to a proinflammatory and pro-restoration phenotype is characterized by a set of critical molecules and intercellular interactions. Given that recruited macrophages play an essential role in coordinating liver repair and regeneration, exogenous, and ex vivo differentiated macrophages with the appropriate phenotype might be used to speed up liver injury recovery [92].

### 3.3. Dendritic Cells (DCs) in AILI

Dendritic cells (DCs), as the most important antigen-presenting cells (APC), are a critical link between innate and adaptive immunity [93]. According to the different differentiation pathways, human DCs originated from hematopoietic stem cells, and divided into myeloid DCs (mDCs) and plasmacytoid DCs(pDCs) [94]. Even though most DCs are immature in hepatic, a small part of CD11c^hi^ conventional DCs (cDCs) express high levels of costimulatory molecules [95]; therefore, DCs are more commonly mediate tolerance rather than immunogenicity in the liver [96,97]. When the liver is wounded, DCs shift their immunological phenotype and become highly immunogenic, modulating NK cells and activating T cells through the production of TNF-α [98]. In AILI, DCs reveal a protective role. Connolly et al. found that liver DCs immune-phenotype was markedly altered after APAP treatment, which expressed higher MHC II, costimulatory molecules, and TLR, and produced IL-6, monocyte chemoattractant protein-1 (MCP-1), and TNF-α [68]. In the AILI model, depleted DCs aggravated liver necrosis and increased mouse mortality; on the contrary, endogenous DCs expansion using FMS-like tyrosine kinase 3 ligand (Flt3L) blocked AILI progression in mice. The protective mechanism may prevent NKs cell activation and induce neutrophil apoptosis [68]. Due to the role of DCs in AILI being less studied, the dual of DCS needs to be further investigated.

### 3.4. Natural Killer Cells (NK Cells) and NKT Cells in AILI

Innate lymphocytes, such as NK cells, NKT cells, and γδT cells, as well as adaptive lymphocytes, such as αβT cells and B cells, are stored in the liver. NK cells, NKT cells, and T cells are up to 65% of all hepatic lymphocytes in humans [99], with NK cells accounting for 30–50% [100]. The liver NK cells can divide into two subsets, CD49a^−^DX5^+^ (conventional NK cells, cNK cells) and CD49a^+^DX5^−^ (the liver-resident NK cells) [101]. NKT cells are a subpopulation of T cells that express both NK cell receptors and T cell receptors, and are MHC I–like molecules, are CD1d-restricted, and are glycolipid antigen reactive [102]. Activated NKs and NKTs produce inflammatory mediators such as interferon (IFN)-γ, TNF-α, IL-10, and IL-4, then induce apoptosis. Activated NK cell ligand-protein levels were significantly increased after drug exposure and activated NK cells exacerbated hepatocytotoxicity by secreting IFN-α. In contrast, specific antibodies for NK cell receptors attenuated drug hepatotoxicity [103]. 

It has been reported that activation of NKs and NKTs amplifies the immune response and leads to the exacerbation of AILI by upregulating IFN-γ, FasL-Keratinocyte-derived chemokine (KC), IP-10 (interferon-inducible protein), Mig (monokine induced by IFN-gamma), MCP-1, macrophage inflammatory protein (MIP)-1α, and increased neutrophil accumulation in the liver [41]. Indeed, the pathogenic effects of NKT cells and NK cells in AILI in mice were dependent on the presence of Dimethyl sulfoxide (DMSO), and DMSO-activated hepatic NKT cells and NK cells in vivo, as evidenced by increased numbers of NKT cells and increased intracellular levels of the cytotoxic effector molecules IFN-γ and granzyme B [69]. Depletion of NK cells and NKT cells with NK1.1 antibodies attenuate AILI [41]. Similarly, Vα14iNKT-cells-deficient (Jα18^−/−^) mice could be resistant to AILI, possibly related to APAP metabolic alterations resulting in reduced hepatic GSH binding and glucuronide binding [70]. 

However, studies have also shown that NK/NKT cells have a protective effect against AILI. For example, NKT-cells-deficient mice (CD1d^−/−^ and Jα18^−/−^) were more vulnerable to AILI than wild-type animals, by a mechanism that involves upregulation and activation of CYP2E1 expression, ultimately leading to APAP protein adduct formation [71]. NKT cells also mediate endogenous IL-4 production, while glutathione synthesis is regulated by endogenous IL-4 under stress conditions to control the severity of AILI [71]. Kwon et al. also found that NKT cells are beneficial in AILI, but they also reduce the release of inflammatory cytokines [72]. Currently, there is no definitive research on the effect of NK/NKT cells in AILI.

### 3.5. γδT Cells in AILI

γδT cells are another type of lymphocyte that plays an essential role in immune response and immunopathological processes and have received widespread attention [104,105]. Like NKT cells, γδT cells are innate-like T cells, which are an integral component of innate immunity and play an essential role in killing infected or damaged cells and regulating functions of innate cells [106]. It was found that IL-17A^+^ CD3^+^ γδT cell receptor (TCR) (+) cells were significantly increased in the AILI model, and depletion of γδT cells significantly reduced IL-17A production and attenuated liver injury by reducing hepatic neutrophil recruitment; further studies revealed that γδT cells were activated by macrophage-derived IL-1β and IL-23 [34]. Hypoxia-inducible factor-1 (HIF) expression in T cells attenuated abnormal γδT cell recruitment and alleviated APAP-induced acute inflammatory response, thereby reducing neutrophil infiltration in the liver [73]. Both studies suggest that γδT cells play a traumatic role in AILI; however, the specific mechanism of injury remains to be further studied.

### 3.6. Cytokine Storm in AILI

Cytokine storm was first described in 1993 [107] that developed after chimeric antigen receptor (CAR) T-cell therapy [108]. David C. illustrates that cytokine storm and cytokine release syndrome (CRS) were life-threatening systemic inflammatory syndromes involving elevated levels of circulating cytokines and immunity caused by various therapies, pathogens, cancer, autoimmune diseases, and single-gene diseases [109]. There is no doubt that the changes in serum cytokines in AILI patients and animal models may contribute to the development of systemic inflammation and may even lead to acute liver failure or MOF.

In AILI, necrotic hepatocytes release DMAPs to recruit immune cells, subsequently activating immune cells to release cytokines (TNF-α, IL-1β, IL-6) and chemokines (MCP-1). The released cytokines then are involved in inflammatory responses or immune-mediated liver injury, resulting in a vicious cycle between cytokines and liver damage. We know that innate immune cells play a dual role in AILI, so the cytokines released by immune cells may play different roles in different studies. For an instant, TNF-α, IL-1β, and IL-6 were reported to aggravate liver injury in AILI. However, some studies implied that TNF-α, IL-6, IL-1β, and IL-8 were crucial cytokines in promoting hepatocyte restoration, this dual role has been discussed in another review [49]. We will discuss IL-1β further in the inflammasome section. There are some cytokines with defined roles reported in the literature. For example, IL-10 [110], IL-4 [111], and IL-13 [112] promote liver tissue repair, and IL-17 [113], IL-8 [114], IL-18 [115], and IFN-γ [110] secretion contribute to the progression of liver inflammation response in APAP challenge.

Why do cytokines appear to have opposite results in AILI? First, the mechanism of cytokine secretion is complex, and the same cytokine may be secreted by several immune cells. For example, TNF-α can be secreted by KCs, macrophages, and DC cells; the mechanism of KCs in AILI is oppositional [52,54]. Second, cytokine secretion is also regulated by cellular signaling pathways, such as JNK [116], STAT3 [117], MAPK [118], and TLR4 [50] signaling pathways. Our previous study suggested that the notch pathway reduces macrophage and neutrophil infiltration to alleviate AILI, reducing mRNA expression with TNF-α, L-1β, and IL-6 [119]. The crosstalk between different signals can affect the expression of cytokines. Third, the protocol of the experimental trials may influence the other cytokine expressions in the literature; for example, serum IL-6, TNF-a, and IL-10 levels are higher in APAP-induced hepatic failure than in APAP-induced hepatic injury using RNA sequencing [120]. In the end, due to the short half-lives of cytokines, serum levels of cytokines may not accurately reflect the ranks in local tissues, and the assay conditions are not consistent in different labs, so there may be discrepancies reported the literature [109]. Other factors should also be considered, such as cytokine detection time points, animal strain, intervention methods, etc.

Lastly, there may be inconsistencies between animal models and patients with APAP overdose. For instance, in animal experiments, IL-10-deficiency-enhanced cytokine secretion (e.g., TNF-α, IL-1α) and iNOS expression have been observed [121]. At the same time, a minor increase in plasma IL-10 has been observed in patients with APAP overdose, and there was no relationship between IL-10 concentrations and the severity of the hepatic injury [122]. Therefore, the mechanisms of cytokine secretion, regulation, and effects are complex and need to be further investigated. 

## 4. Inflammasomes in AILI

Inflammasomes are mainly multimeric protein complexes of sensor, adaptor, and pro-caspase-1 components, such as NACHT-, LRR-, and PYD-domain-containing proteins (NALPs); apoptosis-associated speck-like proteins which contain a CARD(ASC); and pro-caspase 1(CASP1) [123]. The inflammasome is the receptor of innate immune cells, which detect circulating DAMPs and PAMPs through the inflammasome, then participate in the innate immune response by activating caspase-1 to cleave pro-Ilβ and pro-IL-18 into IL-1β and IL-18 [124] (Figure 1). For instance, IL-1β recruits immune cells and causes the programmed death of cells by binding to IL-1R-expressing immune cells; whether this mode of cell death is necrosis or pyroptosis remains further explored [22]. DAMPs activate inflammasomes via TLRs, causing transcriptional activation of proinflammatory cytokines genes in neutrophils and macrophages, mediating sterile inflammation in the liver. However, it has also been suggested that the primary purpose of the inflammasome in sterile inflammation is to remove necrotic cellular debris to make room for hepatocyte proliferation and promote liver tissue regeneration [22].

The role of the inflammasome in AILI remains controversial. Some studies suggest that inflammasome can promote AILI. Imaeda et al. identified the role of TLR9 and NALP3 inflammasomes in promoting AILI using TLR9-deficient (*Tlr*^−/−^) and NALP3-deficient mice (*Casp1*^−/−^, *ASC*^−/−^, *Nalp3*^−/−^) [115]. They discovered that APAP causes hepatocyte death and that free DNA produced by apoptotic hepatocytes activates TLR9, then triggers a signaling cascade in sinusoidal endothelial cells, increasing transcription of pro-IL-1β and pro-IL-18 genes. The NALP3 inflammasome plays a critical role in the second stage of proinflammatory cytokine activation following AILI by activating caspase-1, the enzyme responsible for producing mature IL-1β and IL-18 from pro-IL-1β and pro-IL-18, respectively [115]. This mechanism for attenuating AILI may be through blocking NALP3 inflammasome formation, reducing the pro-IL-1β expression, diminishing serum IL-1β and decreasing Gr1^+^ neutrophil infiltration. Furthermore, IL-1β requires binding to IL-1R-expressing immune cells to achieve biological functions. It was shown that IL-1R-deficient (IL1R^−/−^) mice were protected against AILI with histological evidence of damage was reduced, although this study showed antibodies against IL-1β and IL-1R similarly attenuated the hepatotoxicity of APAP [125]. Another study supports this result by using IL-Rα-deficient (IL1Rα^−/−^) mice to protect against APAP overdose, showing that NLRP3 inflammasome activation of IL-1β is unessential to AILI [126]. Other studies using recombinant human IL-1R antagonist (rhIL-1Ra) significantly improved the survival rate of AILI mice [127]. DAMPs activate inflammasome via TLRs. Studies have shown that benzyl alcohol (BA) can treat AILI by blocking APAP-induced inflammasome signaling, and this effect depends on TLR4 signaling but not TLR2 or CD14. The protective against APAP mechanism of BA through the specific expression of TLR4 in myeloid cells (LyzCre-tlr4^−/−^) [128]. The above studies demonstrate the corrupt effect of the inflammasome in AILI through different mechanisms.

However, IL-1R-deficient mice are not protected against APAP hepatotoxicity [129]. According to previous research, the purinergic receptor antagonist A438079 protects against APAP-induced liver damage by blocking the activation of the NALP3 inflammasome in KCs, preventing inflammation [130]. After APAP overdose, A438079 dramatically reduced GSH depletion, resulting in a 50% reduction in total liver and mitochondrial protein adducts; nevertheless, A438079 suppressed hepatic P450 enzyme activity in a dose-dependent manner without involving the NALP3 inflammasome [131].

The role of the inflammasome in AILI is debated, and different studies in the literature show opposing results. Firstly, the design of the experimental protocol and the reproducibility of experimental animals and reagents may be relevant. Secondly, the composition of the inflammasome is complex, and the downstream activation of effector components needs further study. Lastly, the innate immune regulation in vivo is complex; the interaction between immune cells and the crosstalk between inflammasomes and immune cells remains further investigated [132].

## 5. Other Immune Cells in AILI

Eosinophils are terminally differentiated and highly granulated shape myeloid cells, which secrete cytokines and enzymes to host cells [133]. In APAP overdose patients, eosinophils have been detected in liver biopsies [134]. Further research found that eosinophil-derived IL-4/IL-13 is responsible for the hepatoprotective effect of eosinophils during AILI. Additionally, the p38MAPK/COX/NF-κB signaling plays a key role in causing eosinophils to secrete IL-4/IL-13 in response to IL-33 [135].

In addition to innate immune cells, adaptive immune cells are also involved in DILI [33]. CD8^+^ T cells, also known as cytotoxic T lymphocytes (CTL) or effector T cells, most commonly infiltrate immune cells type in DILI [33]. The proportion of CD4^+^ T cells in the peripheral blood of patients with ALF was higher than the control group [136], suggesting that T lymphocytes play a key role in liver injury. Further research found that T cells exacerbate APAP hepatotoxicity by generating IFN-γ and TNF, enhancing STAT1 activation, and reducing STAT3 activity in the hepatocyte [137].

As we know, naïve CD4^+^ T cells are divided into several subsets, including Th1, Th2, Th17, and regulatory T cells (Treg), based on their unique cytokine production profiles [138]. Following APAP exposure, more CD62L^low^CD44^hi^CD4^+^ T cells (Th1) were infiltrated in the liver, and concomitant IFN-γ was elevated. The reduction in CD4^+^ T cells caused by either genetic insufficiency or antibody depletion significantly weakened proinflammatory cytokine levels and reduced liver damage. While the adoptive transfer of Treg cells reduced APAP-induced liver damage, Treg cell depletion enhanced hepatic CD62L^low^CD44^hi^CD4^+^ T cells, elevated proinflammatory cytokines, and aggravated liver injury [139], suggesting that Treg could ameliorate AILI via secretion of IL-10 and TGF-β. The Th 17 cells were discovered to be the majority of the IL-17-producing cells, and they multiplied quickly after receiving the APAP challenge [140].

## 6. Conclusions and Research Perspectives

The innate immunopathogenesis of AILI is complex, and includes a cascade of reactions involving a wide range of innate immune cells, inflammatory mediators, inflammasomes, and signaling transduction pathways. APAP-induced hepatocyte necrosis triggers a sterile inflammatory response. Still, the mechanism of cell death remains unclear, including the combined involvement of multiple agencies such as increased mitochondrial permeability, endoplasmic reticulum stress, and oxidative stress to cell death. The means of cell death, including necrosis, apoptosis, and pyroptosis, remain to be further investigated. However, the mechanism by which hepatocyte death releases various DAMPs, recognizes immune cells through pattern recognition receptors, and secretes cytokines and chemokines—recruiting neutrophils and monocytes into the liver and initiating the immune cascade response—is indisputable. Immune cells play a dual role in AILI; as described in our study, neutrophils, KCs, macrophages, NK/NKT cells, and inflammasomes can promote liver injury or regeneration. Thus, the immune response’s proinflammatory and pro-repair of the liver needs further investigation.

The proinflammatory and pro-regenerative debate is related to the complexity of the immune response. This involves cytokines being released by immune cells in different microenvironments, switching in cellular manifestation leading to altered functions, mutual crosstalk between immune cells, and crosstalk related to immune cells and inflammatory factors. Secondly, it is related to the experimental protocol, the dose of APAP used in different experiments, the APAP administration regimen (intravenous, intraperitoneal, oral), the experimental animal strain, and the in vitro practical cell line [49]. Thirdly, inconsistencies in how immune cells and inflammatory factors intervene in the experiment, such as inhibitors, antibodies, and gene deletions (complete knockout, specific knockout, mutations), may lead to biased results.

In this study, we retrospectively reviewed the innate immune mechanisms of APAP-induced liver injury, demonstrated the proinflammatory and pro-restorative agents of immune cells, inflammasomes, and cytokines in AILI, and analyzed the possible reasons for the contrasting results. Therefore, in future studies, we should consider the overall experimental protocol design and the reliability and reproducibility of the experimental results to understand the innate immune mechanisms in AILI.

## Figures and Tables

**Figure 1 biology-11-01057-f001:**
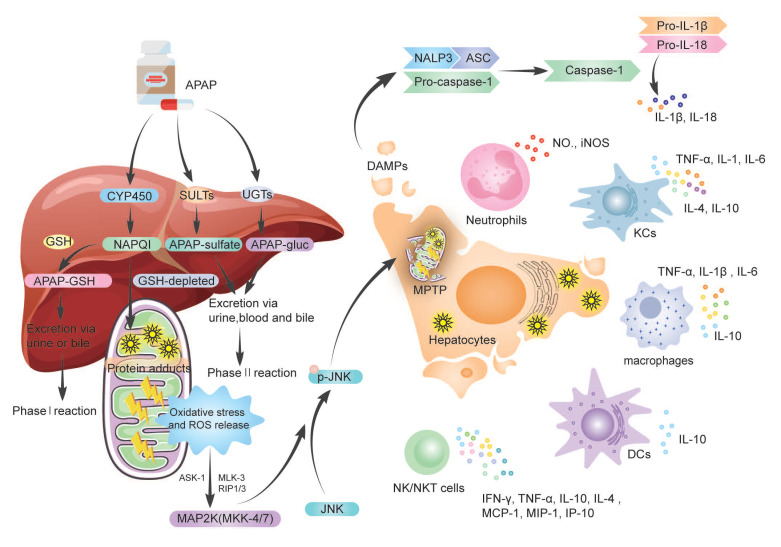
The metabolism of APAP and the role of innate immune response in AILI. Following oral treatment, APAP is absorbed from the gut and transported to the liver for metabolism. A large portion (80–90%) of APAP is metabolized by SULTs and UGTs. A minor (5–10%) amount of APAP was metabolized in hepatocytes by CYP450 enzymes to the reactive metabolite NAPQI. The GSH rapidly converts NAPQI forming the APAP-GSH complex. When GSH is depleted, the growing concentration of NAPQI forms harmful APAP protein adducts, resulting in oxidative stress and increased ROS production in the mitochondria of hepatocytes. Protein adducts induce phosphorylation of JNK via ASK-1, MLK-3, and RIP1/3, then induce the MPTP opening, ultimately leading to hepatocyte necrosis and liver failure. The necrotic hepatocyte released various endogenous DAMPs, upregulating the infiltration of neutrophils, monocytes/macrophages, activated KCs, DCs, NK/NKT cells, and secreted cytokines such as TNF-α, IL-1, IL-6, IL-10, etc. DAMPs also activate inflammasomes, which participate in the innate immune response by activating caspase-1 to cleave pro-ILβ, and pro-IL-18 into IL-1β and IL-18. APAP—acetyl-para-aminophenol; AILI—APAP-induced liver injury; SULTs—sulfate transferase; UGTs—UDP glucuronosyltransferase; NAPQI—N-acetyl-p-benzoquinone imine; ROS—reactive oxygen species; MAPK—mitogen-activated protein kinase; JNK—c-Jun N-terminal kinase; MPTP—mitochondrial membrane permeability transition pore; DAMPs—damage-associated molecular patterns; KCs—Kupffer cells; DCs—Dendritic cells; NK/NKT cells—natural killer cells/NKT cells; TNF-α—tumor necrosis factor-α; IL—interleukin; INF—interferon; MCP—monocyte chemoattractant protein; MIP—macrophage inflammatory protein; IP—interferon-inducible protein; GSH—glutathione; CYP450—cytochrome P450; MLK-3—maxed-lineage kinase-3; ASK-1—apoptosis signal-regulating kinase-1; MKK—MAPK kinases; RIP1/3—receptor-interacting protein-1/3.

## Data Availability

Not applicable.

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
