# Peer review of "The Dual Role of Innate Immune Response in Acetaminophen-Induced Liver Injury"

_biology, 2022, doi:10.3390/biology11071057_

Round 1

Reviewer 1 Report

The Review manuscript titled "The dual role of innate immune response in Acetaminophen- 2 induce liver injury" has been written in detail and will be of broad interest to the readers. Below are my comments:

Major comment:

The dual roles of the immune cells (inflammatory and regenerative) need to be outlined more clearly. Perhaps in the respective sections that describe each cell type, there should be two columns with two roles (inflammatory and regenerative) with references.

Minor comments:

1. Please write acetyl-para-aminophenol when writing the abbreviation as APAP.

2. In the legend of Figure 1, full forms of all the abbreviations should be mentioned. In the figure, JNK leads to p-JNK. Not p-JNK to p-JNK.

3. In the first paragraph of the Introduction, please mention acute injury versus chronic injury. Please also mention the dose (eg. mg/day) for the acute and chronic liver toxicity.

4. For the dual role of immune cells and cytokine (section 3) please add a table with two roles in 2 columns and references.

5. In the section 3.1 it says "The study revealed that the recruited neutrophils (Mac-1+ Gr-1+) significantly increased in the hepatic Hepatic what? should be replaced by "liver".

6. There are words that appear in bold by mistake.

Reviewer 2 Report

Yang et colleagues has written a very comprehensive and thorough study on the complex and controversial role of innate immune response in acetaminophen induced liver injury. The review is very well structured and easy readable.  Most important studies on the field are included and well discussed. I would suggest a minor comment about two recent articles regarding the role of platelet-neutrophil activation in APAP overdose (Chauhan et al Nat comm, 2020) and on the role of eosinophils in AILI (Xu and Yang et al Hepatology, 2022).

Overall, the review is very high quality and I recommend it for publication.

Reviewer 3 Report

 The article ‘The dual role of innate immune response in Acetaminophen-induce liver injury’ by Yang et al. summarizes the current understanding of Acetaminophen-induce liver injury (AILI). The authors further describe the multiple functions of various immune cells and their secreted cytokines in this process. The review article is structured nicely, and the work is relevant for the readership of ‘Biology.’ However, the authors need to address some critical concerns in the current version of the manuscript before its publication. Further, the quality of written language needs to be significantly improved. Hence, the article requires proofreading by a native English speaker. The major and minor issues are listed below.

Major:

Figure1 does not fully explain how protein adduct is responsible for the induction of MAPK. 

It would be helpful for the readers to highlight and distinguish the phase I and II reactions/NAPQI accumulation in Figure1.   

The section ‘Dendritic cells (DCs) in AILI’ only described DCs tangentially without enough discussion on their roles in AILI. 

The dual role of some immune cells, as claimed by the authors (such as in DCs and γδT cells) in AILI, is not discussed convincingly. 

The authors claim that ‘Cytokine storm was first described in 2010 that developed after chimeric antigen receptor (CAR) T-cell therapy [107].’ However, it was coined much before that. Indeed, it was first described in the context of graft-versus-host disease (GVHD) in 1993 (Clark, IA; 2007).

A table summarizing different types of immune cells involved in AILI, their dual roles, and mediators would be convenient for the readers.

Many studies have also highlighted the roles of adaptive immune cells such as CD4 T helper and CD8 cytotoxic T cells in AILI. These cells also seem to have dual roles. It would be a more comprehensive review if the authors included a section describing some of these studies.

There are many verb tense inconsistencies throughout the manuscript. Further, the writing style, typographical and grammatical errors should be corrected in the revised version of the manuscript. For example, the title should read ‘Acetaminophen-induced liver injury’. There are many such errors throughout the paper.  

Minor: 

Abstract, line 22: DC cells? Do the authors mean dendritic cells here? 

Figures are not numbered. 

Page 2, line 32: APAP is abScheme 5. of APAP was metabolized in hepatocytes? 

It would be helpful to expand abbreviations in the figure legend. 

Round 2

Reviewer 3 Report

After careful examination of the revised manuscript, the response of the authors to previous reviews, and the changes made in the manuscript, I gather that the revised version of the manuscript has addressed the major concerns raised in the previous version of the paper (the limitations about unresolved comments are understandable). Hence, I endorse the publication of this paper.